# Metalloporphyrins Reduce Proteinuria in Podocyte Immune Injury: The Role of Metal and Porphyrin Moieties

**DOI:** 10.3390/ijms241612777

**Published:** 2023-08-14

**Authors:** Elias A. Lianos, Gia Nghi Phung, Michelle Foster, Jianping Zhou, Mukut Sharma

**Affiliations:** 1Salem Veterans Affairs Health Care System, Salem, VA 24153, USA; gia.phung@va.gov (G.N.P.); michelle.foster3@va.gov (M.F.); 2Department of Basic Science Education, Virginia Tech Carilion School of Medicine, Roanoke, VA 24016, USA; 3Kansas City VA Medical Center, Kansas City, MO 64128, USA; jianping.zhou@va.gov (J.Z.); mukut.sharma@va.gov (M.S.)

**Keywords:** Heymann nephropathy, heme oxygenase-1, podocytes, proteinuria, iron protoporphyrin, cobalt protoporphyrin, CD55, CD59, complement, complement regulatory proteins

## Abstract

Depending on their central metal atom, metalloporphyrins (MPs) can attenuate or exacerbate the severity of immune-mediated kidney injury, and this has been attributed to the induction or inhibition of heme oxygenase (HO) activity, particularly the inducible isoform (HO-1) of this enzyme. The role of central metal or porphyrin moieties in determining the efficacy of MPs to attenuate injury, as well as mechanisms underlying this effect, have not been assessed. Using an antibody-mediated complement-dependent model of injury directed against rat visceral glomerular epithelial cells (podocytes) and two MPs (FePPIX, CoPPIX) that induce both HO-1 expression and HO enzymatic activity in vivo but differ in their chelated metal, we assessed their efficacy in reducing albuminuria. Podocyte injury was induced using rabbit immune serum raised against the rat podocyte antigen, Fx1A, and containing an anti-Fx1A antibody that activates complement at sites of binding. FePPIX or CoPPIX were injected intraperitoneally (5 mg/kg) 24 h before administration of the anti-Fx1A serum and on days 1, 3, 6, and 10 thereafter. Upon completion of urine collection on day 14, the kidney cortex was obtained for histopathology and isolation of glomeruli, from which total protein extracts were obtained. Target proteins were analyzed by capillary-based separation and immunodetection (Western blot analysis). Both MPs had comparable efficacy in reducing albuminuria in males, but the efficacy of CoPPIX was superior in female rats. The metal-free protoporphyrin, PPIX, had minimal or no effect on urine albumin excretion. CoPPIX was also the most potent MP in inducing glomerular HO-1, reducing complement deposition, and preserving the expression of the complement regulatory protein (CRP) CD55 but not that of CD59, the expression of which was reduced by both MPs. These observations demonstrate that the metal moiety of HO-1-inducing MPs plays an important role in reducing proteinuria via mechanisms involving reduced complement deposition and independently of an effect on CRPs.

## 1. Introduction

The metal atom chelated at the center of porphyrins (heterocyclic organic compounds consisting of four pyrroles) determines functionality of various metalloporphyrins (MP) and plays a key role in their biological and metabolic significance. In higher organisms, the chelated metal is usually iron (Fe), and the porphyrin is protoporphyrin IX, thus forming the naturally occurring MP, heme (Fe-Protoporphyrin IX, FePPIX). In physiological systems, heme is a key constituent of specific proteins, which either bind oxygen, as in hemoglobin, or function as components of membrane-bound electron transport chains. Heme is also a constituent in the superfamily of cytochromes, in which it plays an important role in supporting oxidative, peroxidative, and reductive metabolism of endogenous and xenobiotic substrates [1]. The ability of heme proteins to carry out these functions is a property of the chelated central metal ion, which must be a transition element capable of undergoing reversible changes in oxidation state (for example, Fe^2+^ ↔ Fe^3+^). The porphyrin moiety enhances the catalytic activity of the metal, and this activity is further augmented by the complexing of the porphyrin with its apoprotein moiety, for example, globin, in the case of FePPIX. 

The direct involvement of FePPIX (heme) in enzyme function has prompted substitution with protoporphyrins containing other metals (Co, Zn, Sn, Cr, etc.) and assessment of this substitution on enzyme function. This approach was further supported by the demonstration that exogenous administration of not only FePPIX but also non-iron PPs, such as cobalt (Co) PPIX, can be incorporated into human membrane cytochrome P450 enzymes without significant perturbation in the overall protein structure [2]. These observations lead to the engineering of new types of metalloporphyrin-based biocatalysts that can perform desired reactivity and effect. 

Depending on their chelated metal atom, MPs can induce or inhibit the synthesis and enzymatic activity of heme oxygenase (HO), which catalyzes the degradation of FePPIX to biliverdin, ferrous iron (Fe^2+^), and carbon monoxide (CO). HO-1 and HO-2 are the two major HO isoforms identified in mammalian cells. While HO-2 is constitutively expressed in basal levels in most tissues, HO-1 is the inducible HO isoform, whose expression is highly increased following exposure to different types of stress and was shown to have antioxidant, cytoprotective, and immunoregulatory effects. MPs capable of inducing HO-1 in vivo (i.e., FePPIX, CoPPIX) have been viewed as therapeutic agents in situations where induction of this HO isoform is desired. 

Consideration of HO-1 induction is based on established broad anti-inflammatory and immunomodulatory effects consequent to FePPIX degradation and generation of cytoprotective metabolites (mainly biliverdin and CO) [3] or other mechanisms that are independent of these metabolites [4]. Both FePPIX and non-Fe MPs have been used in clinical practice or clinical trials. For example, FePPIX (available for clinical use as hematin) remains the only specific treatment and a first-line drug in the treatment of acute porphyria [5], while other non-Fe MPs have been proposed for the treatment of disorders, including ischemic injury [6], neonatal hyperbilirubinemia [7], and neutropenia [8]. Furthermore, the metal-devoid PPIX, which is the immediate biosynthetic precursor of FePPIX, has been used in photodynamic therapy of specific skin disorders and tumors [9].

Studies presented here employed an antibody-mediated complement-dependent rat model of glomerular injury directed against visceral glomerular epithelial cells known as Heymann Nephropathy (HN) and resembling human forms of membranous nephropathy [10]. In this model, we assessed the efficacy in reducing proteinuria of two HO-1-inducing metalloporphyrins (FePPIX and CoPPIX) that differ in their central metal moiety as well as the efficacy of the metal-free PPIX.

## 2. Results

### 2.1. Effect of MP Treatment of Clinical Phenotype

MP treatment had no apparent effect on locomotor activity and did not cause an anxiety-like behavior in male or female rats. Both female and male rats with Passive Heymann Nephropathy (PHN) induced by the administration of the anti-Fx1A antibody (Ab group in Figure 1a,b) gained weight during the 14-day study period. However, this gain was significantly attenuated compared to the control groups. Both female and male rats with PHN treated with CoPPIX failed to gain weight. In female rats with PHN, the FePPIX treatment had no effect on weight gain, but the PPIX treatment significantly reduced this gain (Figure 1a). In male rats with PHN, neither the FePPIX nor PPIX treatment had an effect on weight gain (Figure 1b).

### 2.2. Metalloporphyrin (MP) Treatment Reduces Proteinuria in Rats with Passive Heymann Nephropathy (PHN)

Urine albumin excretion was increased in all animals at the end-point of the study (day 14) following a single intravenous injection of the anti-Fx1A antibody. In glomeruli of rats with albuminuria, there was increased deposition of the mesangial matrix and segmental hypercellularity (Figure 2b) compared to glomeruli of the control rats (Figure 2a). In a number of glomeruli, there was a detachment of podocytes from the glomerular capillary wall (Figure 2c, arrows) and edematous podocytes with enlarged nuclei (Figure 2d, arrows).

Urine albumin excretion, expressed as the albumin-to-creatinine ratio (ACR, mg/g), in MP-treated rats on day 14 following a single intravenous injection of the rabbit anti-rat Fx1A antibody (immune serum) to induce PHN and in control rats (injected with non-immune rabbit serum) are shown in Figure 3. In both female and male rats, anti-Fx1A antibody (Ab) administration significantly increased ACR compared to the control (*p* < 0.0001). Both FePPIX and CoPPIX treatment of animals injected with the anti-Fx1A antibody reduced ACR compared to animals injected with the antibody alone (*p* = 0.04). This effect was more pronounced in the CoPPIX-treated female rats (Figure 3, left panel), in which ACR was reduced to a level no different (*p* = 0.27) from the control. In male rats (Figure 3, right panel), both the FePPIX and CoPPIX treatment significantly reduced ACR to a similar extent, but ACR remained significantly higher (*p* < 0.001) compared to the group treated with anti-Fx1A Ab alone. 

### 2.3. Effect of MP Treatment on Glomerular HO-1 Induction and on Determinants of Immune Injury (Complement Activation and Expression of Complement Regulatory Proteins) in PHN

Following binding of the rabbit anti-rat Fx1A antibody (RbIgG) to glomerular visceral epithelial cells (podocytes), there is complement activation and assembly of complement proteins shown to cause cell injury and increased permeability of the glomerular capillary barrier to protein [10]. Complement proteins C3 and C5b-9 (membrane attack complex, MAC) are invariably present in a distribution similar to that of anti-Fx1A antibody binding, while the complement regulatory proteins (CRP) CD55, also known as decay-accelerating factor (DAF), and CD59 exert inhibitory effects on complement activation. CD55 inhibits the C3/C5 convertase enzymes of the classical and alternative activation pathways. CD59 inhibits the formation of the membrane attack complex (MAC) [11]. HO-1 overexpression in podocytes was shown to increase CD55 expression and reduce C3 deposition in immune-mediated glomerular injury [12]. We, therefore, examined the effect of treatment with the HO-1-inducing MPs, FePPIX and CoPPIX, on glomerular HO-1 induction, complement deposition, and CRP (CD55, CD59) expression. 

#### Validation of HO-1, CD55, and Complement Protein Detection in Normal Rat Glomeruli

In normal rat glomeruli, FePPIX (heme) dose-dependently induces HO-1 [13]. We, therefore, assessed the reproducibility of HO-1 induction using the anti-HO-1 polyclonal antibody employed in this study (raised against HO-1/HMox1 fusion protein Ag1190) (see Section 4). HO-1 induction in protein lysates of normal rat glomeruli incubated in serum-free media for 4 h in the absence or presence of heme (hemin 0, 5, and 10 µM) is shown in Figure 4. HO-1 (51 kDa) was detected in the absence of hemin (0 µM) and was induced by 5 and 10 µM hemin.

In normal rat glomeruli, CD55 expression is restricted in podocytes [14]. We, therefore, assessed CD55 protein detection using the anti-rat CD55 polyclonal antibody employed in the present study (raised against rat CD55 C-terminus and corresponding to 250aa of this terminus) (see Section 4). CD55 protein (93 kDa) in lysates of separate preparations of isolated normal glomeruli (*n* = 12) cultured for 4 h is shown in Figure 5. CD55 was also detected in cell-free culture media in which a higher molecular weight CD55 isoform (127 kDa) was also present. 

Constituent cells of normal rat glomeruli, including mesangial cells and podocytes, were shown to synthesize C3, which is an essential component of both the classical and alternative pathways of complement activation, and C4, an essential component of the classical (antibody-mediated) pathway [15]. This was verified in protein lysates of normal rat glomeruli following a 4- or 18-h culture in serum-free media using a rabbit monoclonal antibody against recombinant C3/C3b. C3b alpha chain (103 kDa) detection in lysates of glomeruli (*n* = 9 preparations) incubated for 4 h is shown in Figure 6a. In lysates of glomeruli incubated for 18 h, two additional complement proteins (55 and 48 kDa), cross-reactive with the anti-C3/C3b antibody employed, were detected (Figure 6b). 

### 2.4. Differential HO-1 Induction in Glomeruli by MP Treatment

Figure 7 shows rabbit IgG (RbIgG) levels (originating from the intravenous injection of rabbit anti-rat Fx1A immune serum) and of HO-1 protein levels in protein lysates of glomeruli isolated from MP-treated female rats on day 14 following a single intravenous injection of anti-Fx1A antibody and treated with MPs employed. Both proteins were assessed by capillary-based separation and immunodetection (Figure 7a) and expressed as the chemiluminescence-based surface area of RbIgG or HO-1 peaks obtained in the electropherogram and factored (corrected) by total lysate protein loaded (Figure 7b). RbIgG was detected in all glomerular protein lysates obtained from anti-Fx1A serum-injected animals, but inter-animal levels varied. There was minimal yet detectable RbIgG in lysates from control animals that received non-immune rabbit serum. Prominent HO-1 protein expression was detected only in glomerular lysates from CoPPIX-treated animals. The level of HO-1 expression was independent of that of RbIgG present in same lysate.

In Figure 8, the extent of rabbit IgG (RbIgG) deposition and HO-1 expression in protein lysates of glomeruli isolated from male rats that received anti-Fx1A Ab and assigned to metalloporphyrin (MP) treatment groups. Similar to female rats (Figure 7), Rb IgG was detected in all glomerular protein lysates obtained from anti-Fx1A Ab-injected animals. Likewise, prominent HO-1 protein expression was detected only in glomerular lysates obtained from CoPPIX-treated male rats, and the level of HO-1 expression was independent of the extent of RbIgG deposition.

### 2.5. Effect of MP Treatment on Complement Deposition and CRP Expression

Following the binding of anti-Fx1A antibody (RbIgG) to glomerular visceral epithelial cells (podocytes), there is complement activation and assembly of complement proteins shown to account for the development of proteinuria (a marker of podocyte injury) [10]. Complement C3 protein and the C5b-9 (membrane attack complex, MAC) are invariably present in a distribution similar to that of anti-Fx1A antibody binding while the complement regulatory proteins CD55, also known as decay-accelerating factor (DAF), and CD59 exert inhibitory effects on complement activation. CD55 inhibits C3/C5 convertase enzymes of the alternative and classical pathways. CD59 inhibits the formation of the membrane attack complex (MAC) by binding of to C8 and preventing the C5b-8 catalyzed unfolding and insertion of multiple C9 molecules into the membrane, thereby blocking cell lysis [11]. HO-1 overexpression in podocytes was shown to increase CD55 expression and reduce C3 deposition in immune-mediated glomerular injury directed against the glomerular basement membrane [12]. Therefore, the observation that MP, particularly CoPPIX, treatment markedly induced glomerular HO-1 (Figure 7 and Figure 8) and reduced albuminuria (Figure 3) raised the question of whether these effects were associated with reduced complement deposition and enhanced CD55 or CD59 expression. This was assessed in glomerular lysates from hemin (FePPIX)-, CoPPIX-, and PPIX-treated rats. Glomerular protein lysates with comparable anti-Fx1A rabbit Ig (the complement activating antibody bound to rat podocytes) or comparable endogenous rat Ig (reflecting the immune response to podocyte-bound rabbit Ig) levels, as determined in Figure 7 or Figure 8, were chosen for these analyses. We assessed lysate levels of the C3b protein, which is the proteolytic cleavage fragment of the central complement component C3 and has been used as a durable cell-bound marker of complement activation [16]. C3b binds covalently to C3b-acceptor sites and is the major ligand to complement receptor type 1 (CR1, CD35), which is present in rat podocytes [17]. 

As shown in Figure 9a, C3b (alpha chain, 66kDa) was the major C3 fragment detected in glomerular protein lysates from rats with anti-Fx1A antibody-mediated podocyte injury. FePPIX/hemin (AbH) and CoPPIX (AbCo) treatment reduced C3b levels. C3b reduction was less prominent in PPIX (AbPP)-treated animals. CD55 was detected in glomerular lysates from control animals (Ctl) (Figure 9b), but levels were reduced in lysates from animals with anti-Fx1A antibody-mediated podocyte injury (Ab). In FePPIX (AbH)- or PPIX (AbPP)-treated animals, CD55 levels remain reduced. In contrast, CD55 levels were preserved in CoPPIX (AbCo)-treated animals. In contrast to CD55, podocyte immune injury (Ab) did not appreciably reduce CD59 levels in glomerular lysates (Figure 9c). However, both FePPIX (hemin, AbH) and CoPPIX (AbCo) treatment significantly reduced CD59. In contrast, CD59 levels were preserved in PPIX-treated animals (AbPP). 

## 3. Discussion

The central metal atom of MPs determines their unique properties on HO enzyme activity and isoform expression. FePPIX, CoPPIX, ZnPPIX, SnPPIX, and SnMP are the most extensively assessed MPs in both human and animal studies. It was demonstrated that their effect on HO isoform expression and enzyme activity can occur in tandem or can be discordant and that these effects vary between in vitro and in vivo systems. For example, although both FePPIX and CoPPIX increase HO-1 synthesis and HO enzyme activity in vivo, CoPPIX inhibits activity in vitro. 

In using MPs to manipulate HO isoform expression/enzyme activity, it is important to consider their effect on enzyme activity (originating from both HO-1 and HO-2) rather than to primarily induce the HO-1 isoform, as well as the ability of HO to oxidatively degrade the MP employed. Only MPs that are degradable by HO, such as FePPIX, can generate the cytoprotective metabolites biliverdin, bilirubin, and CO. Non-Fe MPs, exemplified by Ni, Mn, and Sn-PPIX, do not bind molecular oxygen and cannot be oxidatively degraded by HO. However, by causing a sustained increase in HO activity, non-degradable MPs can cause the breakdown of intracellular free heme. The ensuing heme depletion may impair the function of heme-containing enzymes, such as NADPH and cytochromes, in which heme plays an important role in supporting oxidative, peroxidative, and reductive metabolism of endogenous and xenobiotic substrates [1]. In addition, these MPs can deplete the intracellular glutathione (GSH) pool, which prevents damage to important cellular organelles caused by sources such as reactive oxygen species, free radicals, peroxides, and heavy metals [18]. 

In animal models of kidney injury, MPs that inhibit HO enzyme activity, exemplified by SnPP and ZnPP, were used to demonstrate that the intact function of this enzyme is important to preserve renal function. MPs that augment HO-1 synthesis, exemplified by FePPIX and CoPPIX, were used to demonstrate that this effect attenuates the extent of the injury when coupled with increased HO enzyme activity. These observations were expanded to models of immune-mediated kidney injury involving the renal glomerular microvasculature (glomeruli) or interstitium. Earlier studies demonstrated that FePPIX treatment of rats with an aggressive form of glomerular immune injury, resembling that of human rapidly progressive glomerulonephritis (GN), attenuated proteinuria and preserved renal function [19]. CoPPIX had a similar effect in a rat model of glomerular injury resembling human immune-complex-mediated GN [20]. While these studies identified MPs that attenuate immune injury, the importance of chelated metal in determining MP efficacy and underlying mechanisms was not assessed. The present studies addressed this question using a rat model of an antibody-mediated complement-dependent form of podocyte injury that resembles human membranous nephropathy and is known as Heymann nephropathy (HN).

HN [21] has been best characterized in rats. It was originally induced by immunization of rats with autologous kidney cortex [22], and it was later shown that it can also be induced using a rat kidney cortical extract, referred to as Fx1A, enriched in the disease-initiating (nephritogenic) antigen, which is a 330,000 molecular weight glycoprotein (gp330), now known as megalin, and is localized in microvilli of the brush border of renal proximal tubules and in coated pits of podocyte cell membrane [23,24]. To induce nephropathy, rats are immunized with the Fx1A antigenic complex (active HN). Alternatively, antibodies against Fx1A are injected intravenously (passive HN, PHN). Proteinuria in this model develops predictably within 5 to 6 d after injection and results from podocyte injury induced by anti-Fx1A antibody-mediated complement activation and assembly on podocytes of the complement membrane attack complex, C5b-9, or other complement components participating at earlier stages of the complement activation cascade [25,26,27]. Immunofluorescence microscopy demonstrates typical granular peripheral capillary wall deposition of IgG and C3 that is also observed in human membranous nephropathy. With time, the generation of immune complexes decreases, but proteinuria remains. Nephrin redistribution and dissociation from the podocyte actin cytoskeleton, thereby disrupting the integrity of the slit diaphragm, consequent to complement deposition, is a proposed mechanism of increased permeability of glomerular capillary barrier to protein [28], and a similar process was shown to occur in active human membranous nephropathy [29].

Using the PHN model and two MPs (FePPIX, CoPPIX) capable of inducing both HO-1 expression and HO enzymatic activity in vivo but differing in their chelated metal, we assessed their efficacy in reducing albuminuria. FePPIX is the natural substrate/inducer of HO and is oxidatively degraded to ferrous (Fe^2+^) iron, biliverdin/billirubin, and CO, which contribute to HO-dependent antioxidant and immunomodulatory effects [30]. CoPPIX was also shown to be a substrate for HO, and the product of its oxidative cleavage was identified as the natural bile pigment, biliverdin IXalpha isomer [31]. The observations indicate comparable efficacy in reducing albuminuria in males but superior efficacy of CoPPIX in female rats (Figure 3). Treatment of PHN rats with the metal-free PPIX, which is ineffective in altering HO enzyme activity in vivo [32], had minimal or no statistically significant effect on urine albumin excretion. 

Previous studies documenting the effect of MPs in attenuating the extent of renal immune injury and putative underlying mechanisms were reviewed elsewhere [33]. Systemic FePPIX (heme) treatment was previously shown to attenuate the severity of the immune-mediated renal injury, and the heme degradation product, CO, mimics this effect. In lupus-prone mice, FePPIX levels in serum, kidney, and spleen lymphocytes negatively correlated with levels of proteinuria. Moreover, FePPIX supplementation at 15 mg/kg significantly ameliorated the syndrome of lupus, extended lifespan, reduced proteinuria, and alleviated splenomegaly and lymphadenopathy [34]. In the same model, intraperitoneal injection with 100 µmol/kg FePPIX once a week beginning at 6 weeks of age to 21–24 weeks significantly reduced proteinuria and severity of glomerular lesions and decreased immune deposits in glomeruli. In addition, circulating IgG anti-double-stranded DNA antibody levels were significantly decreased in FePPIX-treated mice. It was concluded that one mechanism whereby HO-1 induction achieved by FePPIX treatment attenuated the severity of lupus nephritis was reduced production of pathogenic autoantibodies. 

The HO reaction is a key, if not the sole, mechanism by which cells can generate carbon monoxide (CO) originating from endogenous heme degradation. Convincing evidence that CO can mimic beneficial outcomes of MPs used to increase HO enzyme activity was provided in the same model of Lupus-prone mice. Daily exposure to compressed CO at a concentration of 250 parts per million (ppm) for 2 h for up to 30 weeks significantly decreased infiltration by activated B220^+^ CD4^−^ CD8^−^ T cells. Moreover, it decreased the extent of glomerular proliferative lesions and proinflammatory cytokine production and delayed the decline in kidney function [35].

However, the beneficial effect of FePPIX and CoPPIX in renal immune injury cannot be solely attributed to HO induction and generation of cellular heme degradation products (biliverdin, CO). Subtle changes in the porphyrin metal center, i.e., Fe versus Co, and ring conformation, are also known to influence agonist versus antagonist actions of protoporphyrin IX. This was shown convincingly using porphyrin-based agents to target specific oncoproteins for cancer therapy. For example, REV-ERBβ, an overexpressed oncoprotein that has been used as a target for cancer treatment, binds FePPIX resulting in decreased proliferation in various cancer cells [36]. Replacement of iron with cobalt at the metal center of protoporphyrin IX changes the ligand from an agonist to an antagonist of REV-ERBβ [37]. Protoporphyrin IX binding to the glomerular microvasculature has not been demonstrated.

In addition to having higher efficacy in reducing albuminuria, CoPPIX was also the most potent in increasing glomerular HO-1 protein levels in both female and male rats (Figure 7 and Figure 8). PPIX treatment had no effect, while that of FePPIX was inconsistent (Figure 8b). CoPPIX treatment also resulted in significant weight loss during the treatment period (Figure 1). This effect is expected given the well-established anorectic effect (transient decrease in food intake and a prolonged loss of body weight) of this MP in rats, mice, chickens, and dogs via effects in the hypothalamus [38]. In previous studies, systemic hemin treatment of rats with glomerular immune injury failed to appreciably induce HO-1 in endogenous glomerular cells. Instead, a robust HO-1 induction immunolocalized in infiltrating macrophages, and the reduction of proteinuria observed was attributed to a functional polarization of these effector cells towards the M2 phenotype as a result of HO-1 induction. [19]. Therefore, the inconsistent increase in HO-1 protein levels in glomerular lysates of FePPIX-treated rats is not surprising, as glomerular infiltration by inflammatory cells in PHN is typically absent. In contrast, CoPPIX treatment consistently increased HO-1 protein content in glomerular lysates of PHN animals treated with this MP (Figure 7 and Figure 8). CoPPIX has a much slower elimination rate (3 days) compared with hemin (10 h), and, four weeks after a single CoPPIX dose, the kidney retains the highest levels of cobalt compared to other organs [39]. In contrast, exogenous hemin rapidly complexes with hemopexin and albumin with subsequent transport to the liver, where it is broken down by HO to biliverdin and CO [40]. These differences in pharmacokinetics between FePPIX and CoPPIX could explain the robust HO-1 induction in glomerular lysates of CoPPIX-treated animals. 

Previous studies demonstrated that cultured rat podocytes synthesize and secrete functional complement component C3 [15]. Moreover, in vivo studies using the PHN model demonstrated that, in addition to promoting C3 deposition, binding of the anti-Fx1A antibody to podocytes increases the synthesis of complement components C3 and C4 by these cells [15,41]. This effect was coincident with the onset and progression of proteinuria and peaked 11 to 14 days following injection of the antibody [15,41]. These observations can account for the detection of C3b in short- and long-term cultures of isolated normal glomeruli (Figure 6) and for increased C3b detection in glomerular protein lysates from rats with PHN (Figure 9a). In PHN rats with comparable deposition of complement-activating rabbit Ig in glomeruli, treatment with FePPIX or CoPPIX decreased C3b, and this effect was most prominent in CoPPIX-treated animals (Figure 9a). CoPPIX treatment also preserved CD55 levels (Figure 9b). This complement-regulatory protein (CRP) prevents the formation and accelerates the decay of C3 convertase, C4b2a, in the classical complement activation pathway, and of C3 convertase, C3bBb, in the alternate pathway. By limiting these amplification convertases of the complement cascade, CD55 attenuates the formation of terminal complement activation components that were mechanistically linked with the development of proteinuria in PHN [25]. In this regard, we previously demonstrated that in rats lacking CD55, C3b deposition in glomeruli increased following the binding of anti-Fx1A antibody to podocytes while proteinuria deteriorated [42]. 

The mechanism of preserved CD55 expression in glomerular protein lysates of CoPPIX but not in those of FePPIX-treated rats with PHN (Figure 9) is unknown and could be attributed to the superior efficacy of CoPPIX to markedly induce HO-1 levels (Figure 7 and Figure 8). This is supported by previous studies showing that podocyte-targeted HO-1 overexpression increased CD55 protein levels [12]. However, mechanisms other than the effects of MP treatment on CRP expression should also be considered, as FePPIX treatment reduced C3b deposition (Figure 9a) even though CD55 levels were not preserved (Figure 9b). Moreover, treatment with either MP reduced CD59, a CRP that limits formation of the C5b-9 membrane attack complex and consequent cytotoxicity, in glomerular protein lysates (Figure 9c). Therefore, the reduction of complement deposition and proteinuria in FePPIX- or CoPPIX-treated animals with PHN cannot be solely attributed to the effects of these MPs on CD55 and CD59. The mechanism underlying the reduction of CD59 in lysates from FePPIX- and CoPPIX-treated animals is unknown but apparently linked to the metal moiety as in metal-free PPIX-treated animals, CD59 levels were not reduced (Figure 9c).

In summary, the present study demonstrated that the metal moiety of HO-1-inducing metalloporphyrins (MPs) plays an important role in reducing proteinuria in a model of antibody-mediated complement-dependent podocyte injury resembling membranous nephropathy. CoPPIX is the most potent MP in inducing HO-1 and in reducing proteinuria and complement deposition. While CoPPIX treatment preserved glomerular CD55 levels, both FePPIX and CoPPIX treatments reduced CD59. Therefore, although the reduction in proteinuria could be mechanistically linked to decreased complement deposition, this effect cannot be mechanistically linked to changes in CRP expression. The mechanism(s), at the molecular level, by which FePPIX and CoPPIX reduce proteinuria following complement-dependent podocyte injury remain to be elucidated. A possible mechanism, supported by existing experimental evidence, is outlined in Figure 10. Complement-dependent podocyte injury disrupts actin cytoskeleton and slit diaphragm integrity resulting in proteinuria [28]. Treatment with HO-inducing MPs causes podocyte HO-1 induction, which maintains actin cytoskeletal and microtubular structures, as shown in kidney epithelial cells [43]. This effect preserves slit diaphragm integrity, thereby reducing proteinuria.

## 4. Materials and Methods

### 4.1. Passive Heymann Nephropathy (PHN)

PHN was induced in male and female Lewis rats (150 to 175 g) by a single intravenous (tail vein) injection of proteinuric doses of rabbit immune serum raised against the rat tubular brush border antigenic complex, Fx1A, isolated according to the method described by Edgington, Glassock, and Dixon [22] and generously provided by Dr. Ashok Singh (Vivastem Laboratories, Lombard, IL, USA). Prior to injection, the immune serum was heat-inactivated at 56 °C for 30 min, centrifuged at 13,000 rpm for 60 min at 4 °C, and filtered. Proteinuric doses of the anti-Fx1A immune serum (1 mL/250 g weight) were determined in pilot experiments to reproducibly induce albuminuria in both male and female rats. All studies were completed on day 14 following a single intraperitoneal injection of the immune serum. To verify albuminuria, a urine sample was collected 48 h before administration of the immune serum and on day 14. Albuminuria was expressed as urine albumin-to-creatinine ratio (ACR, mg albumin/g creatinine) measured using a commercially available rat ACR kit (Abcam, Waltham, MA, USA, Catalog# ab241018).

### 4.2. Metalloporphyrin (MP) Treatment Protocols and Tissue Retrieval

The following MPs (all obtained from MilliporeSigma, Burlington, MA, USA) were used: Hemin (Ferriprotoporphyrin IX, FePPIX) Chloride (Sigma, Cat# H9039), Protoporphyrin IX Cobalt chloride (CoPPIX) (Sigma, Cat# C1900) and metal-free Protoporphyrin IX (PPIX). FePPIX was dissolved in 0.1N NaOH, adjusted to pH 7.4 with 0.1N HCl, and used as a working solution at 1:10 dilution in saline. CoPPIX was dissolved in Dimethylsulfoxide (DMSO) and used as a working solution mixed in saline at a concentration of 5 mg/kg. All MPs were injected intraperitoneally (5 mg/kg) 24 h before administration of the anti-Fx1A serum and on days 1, 3, 6, and 10 thereafter. Animal body weight was measured 24 h before administration of the anti-Fx1A serum and on days 7 and 14 thereafter. Eleven female or male rats were assigned to each control and MP treatment group. Upon completion of urine collection, animals were terminally anesthetized, and a slice of kidney cortex was obtained for routine histopathology studies. The remaining cortex was used for the isolation of glomeruli using an established sieving protocol we previously reported [44]. Total protein extracts from isolated glomeruli were obtained using methods we and others previously reported [13].

### 4.3. Capillary-Based Separation and Immunodetection (Western Blot Analysis) of Glomerular Proteins

A Western protein analyzer (ProteinSimple, Biotechne) that performs protein separation, immunoprobing, chemiluminescence detection, total protein normalization, and data analysis was used. Glomerular protein lysate sample preparation/optimization, sample, and stacking matrix loading, assessment of a linear dynamic range of protein lysates to determine the limit of detection, optimization of dilution of antibodies employed, and assessment of protein loading were performed according to manufacturer’s specifications/guidelines. Protein detection in glomerular lysates was factored (normalized) by the total protein loaded in each capillary. An in-capillary labeling technique was used where proteins were separated and then UV-captured in the capillary wall in a manner similar to capturing target proteins to be assayed. Captured proteins were exposed to a biotin labeling reagent, which allows for recognition by streptavidin-HRP and subsequently detected in a chemiluminescent reaction. A total protein detection kit was used according to the manufacturer’s (ProteinSimple, Biotechne) specifications. 

### 4.4. Antibodies

Antibody (Ab) against rabbit IgG (a monoclonal mouse anti-rabbit IgG Ab, Antibodies-online Inc., Pottstown, PA, USA) was used to detect the presence in glomerular protein lysates of the rabbit anti-rat Fx1A Ig antibody injected to induce PHN. Antibody against rat IgG (a polyclonal rabbit anti-rat IgG Ab, MilliporeSigma, Burlington, MA, USA) was used to detect the presence in protein lysates of endogenous rat IgG generated against the exogenously administered rabbit anti-Fx1A Ig that bound in podocytes. The antibody used to detect complement protein component C3/C3b in glomerular protein lysates was a monoclonal mouse IgM raised against an epitope on C3b and reacting with intact rat C3/C3b. It was obtained from HycultBiotech (Clone 2B10b9b2). Ab used to detect HO-1 in protein lysates was polyclonal raised against HO-1/HMox1 fusion protein Ag1190. It was obtained from Proteintech (Catalog# 10701-1-AP). The anti-rat CD55 Ab used to detect CD55 protein in lysates was polyclonal raised against a recombinant HO-1 fragment (His-tag) corresponding to 250aa of rat CD55 C-terminus (Abcam, catalog# ab231061). The anti-rat CD55 Ab used to detect CD55 protein in protein lysates was a rabbit anti-rat CD59 polyclonal antibody against recombinant Rat CD59 protein (ThermoFisher, Cat# 117852). 

### 4.5. Data Analysis

Sample numbers are mentioned under the description of Section 2 for each experiment. Data were analyzed using Student’s *t*-test for two group comparisons. A *p*-value < 0.05 was considered significant.

## Figures and Tables

**Figure 1 ijms-24-12777-f001:**
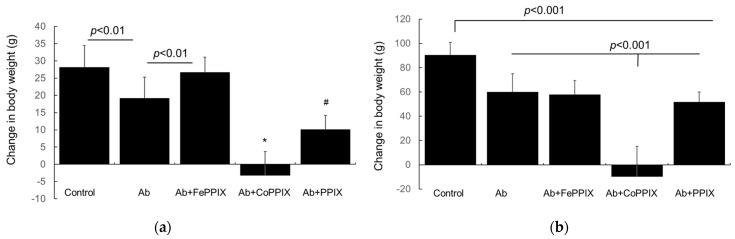
(**a**): Weight gain (mean ± SD, *n* = 11) in female control rats and those that received the anti-Fx1A antibody (Ab) to induce PHN or that received this antibody and were treated with FePPIX (Ab+FePPIX), CoPPIX (Ab+CoPPIX) or PPIX (Ab+PPIX) during the 14-day study period. Weight gain in the control group was not different from that in the Ab+FePPIX group. Weight gain in the control group was significantly greater than that in Ab alone group (*p* < 0.01) and from that in Ab+PPIX (#, *p* < 0.001) and Ab+CoPPIX-treated (*, *p* < 0.001) groups. There was no increase in body weight in the Ab+CoPPIX group. Weight gain in the Ab+CoPPIX-treated group was also significantly less than Ab, Ab+FePPIX, or Ab+PPIX groups (*, *p* < 0.001). (**b**): Weight gain (mean ± SD, *n* = 11) in male control rats and those that received the anti-Fx1A antibody (Ab) or this antibody and treatment with FePPIX (Ab+FePPIX), CoPPIX (Ab+CoPPIX) or PPIX (Ab+PPIX) during the 14-day study period. Weight gain in the control group was significantly higher than in all other groups (*p* < 0.001). Weight gain in groups treated with Ab, Ab+FePPIX, or Ab+PPIX was similar and significantly higher than the group treated with Ab+CoPPIX (*p* < 0.001). There was no increase in body weight in the Ab+CoPPIX group.

**Figure 2 ijms-24-12777-f002:**
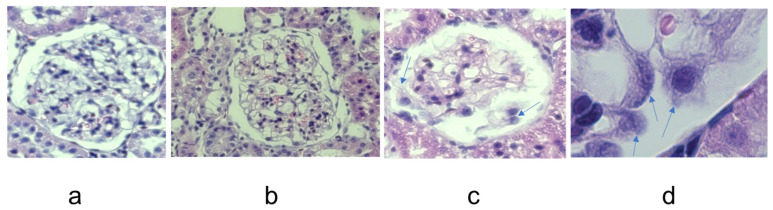
Eosine/hematoxylin stain of 4µm cortical sections obtained from control (**a**) and PHN (**b**–**d**) rats. Increased mesangial matrix and segmental hypercellularity are shown in (**b**). Also shown are detached podocytes ((**c**), arrows) and edematous podocytes with enlarged nucleus ((**d**), arrows). Magnification: 200× (**a**–**c**), 630× (**d**).

**Figure 3 ijms-24-12777-f003:**
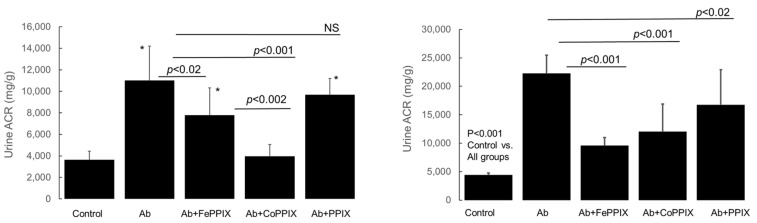
Urine albumin excretion expressed as albumin-to-creatinine ratio (ACR, mg/g) in female (**left panel**) and male (**right panel**) rats on day 14 following a single intravenous injection of rabbit anti-rat Fx1A antibody (immune serum) and in control rats. Effect of metalloporphyrin (MP) treatment on ACR (*n* = 11 in each group). ACR values are expressed as mean ± SD. Ab, group that received anti-Fx1A antibody only; Ab+FePPIX, group that received anti-Fx1A antibody and treated with hemin (FePPIX), as described in Section 4; Ab+CoPPIX, group that received anti-Fx1A antibody and treated with cobalt protoporphyrin (CoPPIX). Ab+PP, group that received anti-Fx1A antibody and treated with metal-free protoporphyrin (PPIX). Statistically significant differences between groups are indicated as *p*-values. Left panel (female rats). *, *p* < 0.001 control vs. Ab, Ab+FePPIX, and Ab+PPIX. FePPIX and CoPPIX treatment reduced proteinuria compared to group that received Ab alone, but CoPPIX was more effective than FePPIX (*p* < 0.001 and 0.02, respectively); NS = not Significant. Right panel (male rats). All treated groups (Ab, Ab+FePPIX, Ab+CoPPIX, and Ab+PPIX) showed significantly higher proteinuria compared to control (*p* < 0.001); Both FePPIX and CoPPIX treatment significantly reduced proteinuria compared to Ab alone group (*p* < 0.001). PPIX treatment also reduced proteinuria to a lesser albeit significant extent (*p* < 0.02).

**Figure 4 ijms-24-12777-f004:**
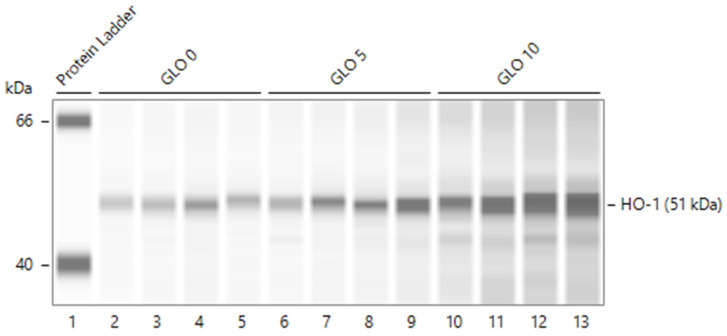
HO-1 induction in isolated normal rat glomeruli incubated in the absence and presence of hemin (0, 5, and 10 µM) in quadruplicate incubation experiments, CLO 0 (capillaries 2–5), incubation in the absence of hemin. GLO 5 (capillaries 6–9), incubations with 5 µM hemin. GLO 10 (capillaries 10–13), incubations with 10µM hemin. Capillary 1, ladder of protein markers.

**Figure 5 ijms-24-12777-f005:**
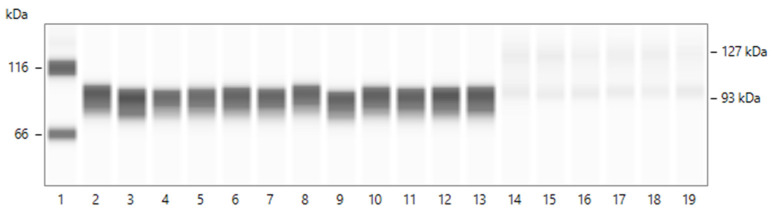
Detection of CD55 protein in lysates of normal rat glomeruli (*n* = 12 separate preparations) cultured for 4 h in serum-free media and in some of these cell-free media as well. A rabbit anti-rat polyclonal antibody raised against a recombinant fragment (His-tag) corresponding to 250aa of rat CD55 C-terminus was used. Capillary 1, ladder of protein standards. Capillaries 2–13, CD55 protein (93 kDa) in lysates of 12 separate glomerular preparations. Capillaries 14–19, detection of CD55 (93 kDa) and of a higher molecular weight (127k Da) isoform in some cell-free culture media of same glomeruli.

**Figure 6 ijms-24-12777-f006:**
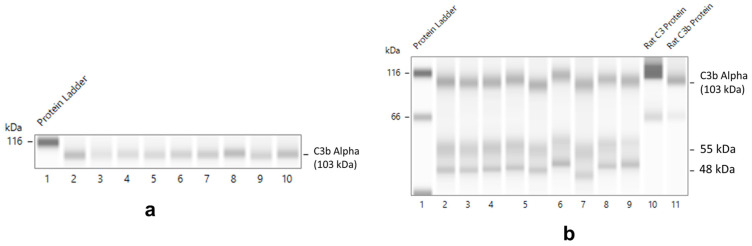
C3/C3b detection in protein lysates of isolated normal rat glomeruli following 4 h ((**a**), capillaries 2–10) or 18h ((**b**), capillaries 2–9) incubation in serum-free culture media using a monoclonal mouse Ab raised against an epitope on C3b and reacting with both intact C3 and C3b (HycultBiotech, Clone 2B10b9b2). Capillaries 10 and 11 in (**b**), authentic rat C3, and C3b proteins, respectively, were used as positive controls. Capillary 1 in (**a**,**b**), protein standard ladder.

**Figure 7 ijms-24-12777-f007:**
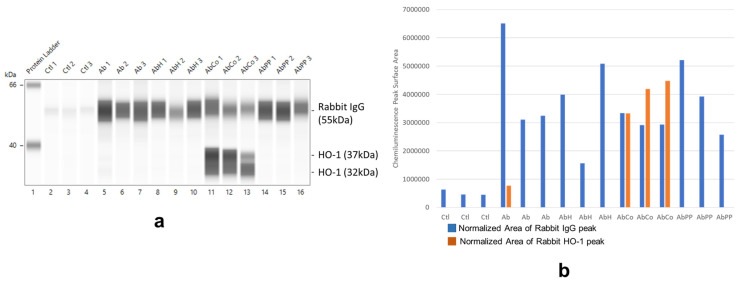
Effect of MP treatment on glomerular HO-1 induction in female rats. Left panel (**a**), capillary electrophoresis. Capillary-based separation and immunodetection of target proteins, rabbit IgG (55 kDa) and HO-1 (32 and 37 kDa), present in total protein lysates obtained from glomeruli of female rats that received anti-Fx1A Ab and assigned to metalloporphyrin (MP) treatment groups. Right panel (**b**), normalization of protein expression. IgG and HO-1 proteins present in each lysate were expressed as the area of RbIgG (blue bars) and HO-1 (orange bars) chemiluminescence peaks obtained in the capillary electropherogram and factored (corrected) by total protein loaded in that capillary. Ctl, control rats (*n* = 3, capillaries 2–4). Ab, rats receiving a single intravenous injection of rabbit anti-rat Fx1A antibody (*n* = 3, capillaries 5–7). AbH, rats receiving the anti-Fx1A antibody (Ab) and treated with FePPIX (hemin, H), as described in Section 4 (*n* = 3, capillaries 8–10). AbCo, rats receiving the anti-Fx1A antibody (Ab) and treated with cobalt protoporphyrin (CoPPIX) (*n* = 3, capillaries 11–13). AbPP, rats receiving the anti-Fx1A antibody (Ab) and treated with metal-free protoporphyrin (PP) (*n* = 3, capillaries 14–16). Capillary 1 in 7a, protein standard ladder. Numbers next to each lysate label indicate the numerical identification of lysate generated for each treatment group (Ctr, Ab, AbH, AbCo, AbPP).

**Figure 8 ijms-24-12777-f008:**
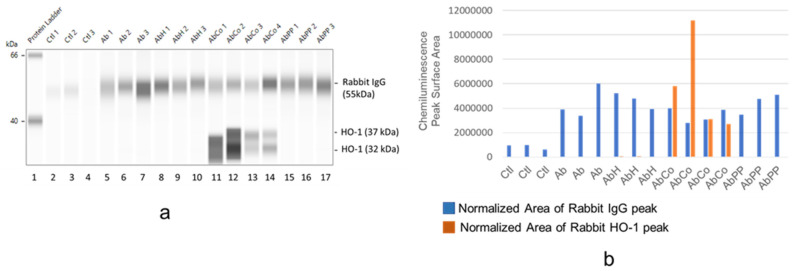
Effect of MP treatment on glomerular HO-1 induction in male rats. Left panel (**a**), capillary electrophoresis. Capillary-based separation and immunodetection of target proteins, RbIgG (55 kDa) and HO-1 (32 and 37 kDa), present in total protein lysates obtained from glomeruli of anti-Fx1A Ab-injected male rats assigned to metalloporphyrin (MP) treatment groups. Right panel (**b**), normalization of protein expression. RbIgG level in each glomerular lysate and HO-1 protein level in same lysate were expressed as a surface area of RbIgG and HO-1 chemiluminescence peaks obtained in the capillary electropherogram factored (corrected) by total protein loaded in that capillary. Ctl, control rats (*n* = 3, capillaries 2–4). Ab, rats receiving a single intravenous injection of rabbit anti-rat Fx1A antibody (*n* = 3, capillaries 5–7). AbH, rats receiving the anti-Fx1A antibody (Ab) and treated with FePPIX (hemin, H), as described in Section 4 (*n* = 3, capillaries 8–10). AbCo, rats receiving the anti-Fx1A antibody (Ab) and treated with cobalt protoporphyrin (CoPPIX) (*n* = 3, capillaries 11–14). AbPP, rats receiving the anti-Fx1A antibody (Ab) and treated with metal-free protoporphyrin (PP) (*n* = 3, capillaries 15–17). Capillary 1 in 8a, protein standard ladder. Numbers next to each lysate label indicate the numerical identification of lysate generated for each treatment group (Ctr, Ab, AbH, AbCo, AbPP).

**Figure 9 ijms-24-12777-f009:**
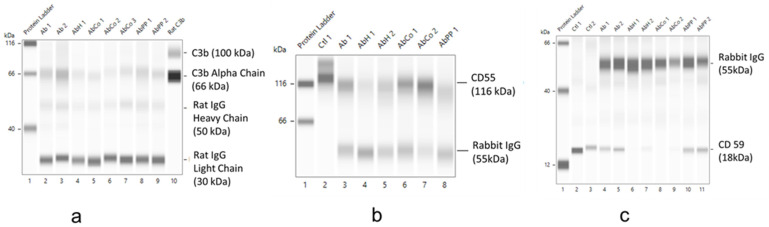
Effect of MP treatment on complement deposition and CRP expression. Capillary-based separation and immunodetection of complement component C3b alpha chain (66 kDa) (**a**) and of complement regulatory proteins CD55 (116 kDa) (**b**) and CD59 (18 kDa) (**c**) present in total protein lysates obtained from control (Ctl) glomeruli and glomeruli of rats with anti-Fx1A antibody (Ab) induced podocyte injury assigned to treatment groups (AbH, AbCo, AbPP). In all three electropherograms, lysate levels of exogenous rabbit Ig (IgG) bound to podocyte Fx1A panels (**b**,**c**) or of endogenous rat IgG generated against podocyte-bound rabbit Ig panel (**a**) are also shown. Ctl, control rats that did not receive anti-Fx1A antibody. Ab, rats receiving a single intravenous injection of rabbit anti-rat Fx1A antibody. AbH, rats receiving the anti-Fx1A antibody (Ab) and treated with FePPIX (hemin, H), as described in Section 4. AbCo, rats receiving the anti-Fx1A antibody (Ab) and treated with cobalt protoporphyrin (CoPPIX). AbPP, rats receiving the anti-Fx1A antibody (Ab) and treated with metal-free protoporphyrin (PPIX). Capillary 1 in all three electropherograms, protein standard ladder. Capillary 10 in (**a**), authentic rat C3b protein standard. Numbers next to each lysate label indicates the numerical identification of lysate generated for each treatment group (Ctr, Ab, AbH, AbCo, AbPP).

**Figure 10 ijms-24-12777-f010:**
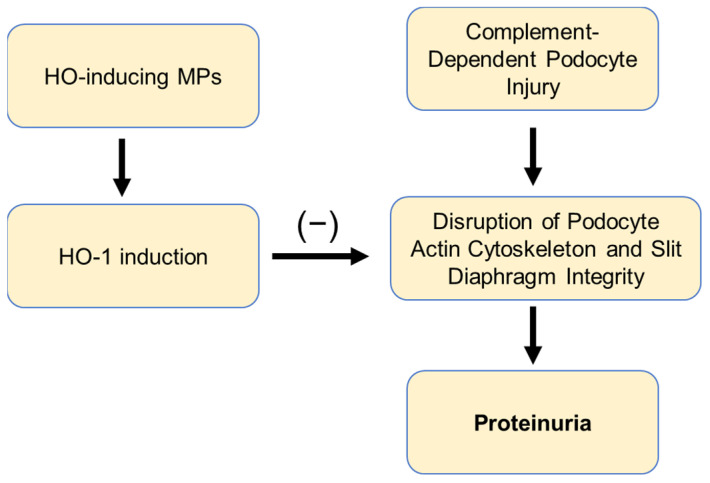
Diagram of proposed mechanism underlying antiproteinuric effect of MP treatment in complement-dependent podocyte injury.

## Data Availability

Not applicable.

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
