# Peer review of "Metalloporphyrins Reduce Proteinuria in Podocyte Immune Injury: The Role of Metal and Porphyrin Moieties"

_ijms, 2023, doi:10.3390/ijms241612777_

Round 1
Reviewer 1 Report
This study provides valuable insights into the role of metalloporphyrins (MPs) in reducing proteinuria in a model of podocyte immune injury. However, there are several aspects of the study that can be improved:
The study briefly mentions previous research on MPs and their effects on immune-mediated kidney injury. However, there is a lack of critical analysis and discussion of the existing literature. A more comprehensive review and comparison of previous findings would strengthen the study's arguments and contribute to the overall understanding of the topic.
Although the study identifies the metal moiety of MPs as an important factor in reducing proteinuria, it does not provide detailed mechanistic insights into how the MPs exert their effects. Further investigation into the underlying molecular pathways and interactions involved would enhance the understanding of the therapeutic potential of MPs.
The study primarily focuses on a rat model of podocyte injury and does not provide insights into the translational potential of the findings to human kidney diseases. Including additional experimental models or human data would increase the relevance and generalizability of the study.
The overall quality of English is good
Author Response
See Word document submitted.
Reviewer 2 Report
The goal of this study is to look at the role of central metal or porhyrin moieties in determining the efficacy of MPs to attenuate injury as well as mechanisms underlying this effect. Actually, the current proposal is interesting and well-written. Therefore, I recommend that the current study be published after minor revisions as follows:
1- Could the authors add a diagrammatic figure to propose the possible mechanistic pathway for these findings?
2- Please add scoring for Figure 2.
3- Please discuss the Metalloporphyrins as Tools for Deciphering the Role of Heme Oxygenase in Renal Immune Injury
Reference; Lianos EA, Detsika MG. Metalloporphyrins as Tools for Deciphering the Role of Heme Oxygenase in Renal Immune Injury. Int J Mol Sci. 2023 Apr 6;24(7):6815. doi: 10.3390/ijms24076815. PMID: 37047787; PMCID: PMC10095062.
Author Response
See Word document submitted.
